# Genomic Resources for the First Federally Endangered Lichen: The Florida Perforate Cladonia (*Cladonia perforata*)

**DOI:** 10.3390/jof9070698

**Published:** 2023-06-24

**Authors:** Steven D. Leavitt, Ann DeBolt, Ethan McQuhae, Jessica L. Allen

**Affiliations:** 1M.L. Bean Life Science Museum and Department of Biology, Brigham Young University, Provo, UT 84602, USA; 2Department of Biology, Boise State University, Boise, ID 83725, USA; annmdebolt@gmail.com; 3Plant and Wildlife Sciences, Brigham Young University, Provo, UT 84602, USA; ethan.mcq01@gmail.com; 4Department of Biology, Eastern Washington University, Cheney, WA 99004, USA; jallen73@ewu.edu

**Keywords:** conservation, Florida scrub, genet, Illumina, MinION, Oxford Nanopore Technologies, ramet, reindeer lichens, transplant

## Abstract

Thirty years after its designation as a federally endangered species, the Florida Perforate Cladonia (FPC) remains imperiled in isolated populations in the Florida scrub in the southeastern USA. For threatened and endangered species, such as FPC, reference genomes provide critical insight into genomic diversity, local adaptations, landscape-level genetics, and phylogenomics. Using high-throughput sequencing, we assemble the first draft nuclear and mitochondrial genomes for the FPC mycobiont—*Cladonia perforata*. We also assess genetic diversity within and among populations in southeastern Florida using genome-scale data and investigate diversity across the entire nuclear ribosomal cistron, including the standard DNA barcoding marker for fungi. The draft nuclear genome spanned 33.6 Mb, and the complete, circular mitochondrial genome was 59 Kb. We also generated the first chloroplast genome, to our knowledge, for the photobiont genus associated with FPC, an undescribed *Asterochloris* species. We inferred the presence of multiple, distinct mycobiont parental genotypes (genets) occurring at local scales in southeastern Florida, and strikingly, no genets were shared among even the closest sample sites. All sampled thalli shared identical mitochondrial genomes, while the nuclear ribosomal cistron showed limited variability—highlighting the genetic resolution provided by nuclear genome-scale datasets. The genomic resources generated here provide critical resources for informed conservation efforts for the FPC.

## 1. Introduction

Florida scrub habitat in the southeastern United States is characterized by well-drained, nutrient-poor sandy soils and supports a diverse assemblage of rare and endemic plant and animal species. In the Florida scrub, unique plant communities are associated with coastal dunes that formed during the Pleistocene, with the scrub vegetation likely remaining intact since the ancient formation of the dunes [1,2,3]. The Florida scrub also supports some of the highest densities of rare, endemic species in the United States, including over two-thirds of Florida’s reptiles and amphibians [4], a number of plants [5,6], arthropods [7], and lichens [8]. This pyrogenic ecosystem is a global biodiversity hotspot where many members of the community are dependent on periodic fire for habitat maintenance [9]. Varying fire return intervals create and maintain open sand gaps where many specialized species occur, including herbaceous plants and lichens [10,11].

Florida scrub habitats are imperiled and in rapid decline, primarily due to anthropogenic activities such as urbanization, agriculture, and land development [12,13], in addition to catastrophic fires and hurricanes [11,14,15]. Furthermore, the Florida scrub is vulnerable to climate change, which is expected to exacerbate the negative impacts of current threats, such as increased frequency and intensity of wildfires, droughts, and hurricanes, further reducing the available habitat for many of the specialized species that rely on this ecosystem [12,16].

The Florida Perforate Cladonia (FPC; mycobiont = *Cladonia perforata* and photobiont = *Asterochloris* spp. [Chlorophyta, Trebouxiophyceae]) is a vagrant lichen occurring on white sandy soils in open areas in sand pine scrub, oak scrub and rosemary bald habitats in the Florida scrub [11,17]. FPC occurs in sixteen disjunct, isolated populations (at ca. 41 sites) from the panhandle area of Florida (North Gulf Coast) to the Lake Wales Ridge in central Florida to Martin and Palm Beach counties in South Florida (Atlantic Coast Ridge) [18] (Figure 1). This lichen likely has a naturally disjunct, patchy distribution, with ongoing local extirpation and recolonization events occurring throughout its long-term persistence in fire-maintained landscapes [11]. Whether these patch-level dynamics generated distinct, locally adapted variants is not known. Presently, habitat patches tend to be small, limited to a few square meters to several hectares, and these sites are exposed to disruptions by hurricanes and fires on very short time scales [11]. Due to significant loss of shrub habitat and limited and patchy occurrences [8,19], in 1993, the FPC was the first lichen to be federally listed as endangered in the U.S.A. and was one of the first lichens to be added to the IUCN Red List [20]. While some populations are relatively large and robust, others are in decline or under threat [21]. The main threats include habitat loss in high-value real estate areas, improper fire management strategies, and impacts from hurricanes.

FPC has a life history suggesting that this lichen reproduces exclusively via thallus fragmentation and clonal growth [21]. Spore-producing structures—apothecia—have not been observed in FPC, and the dispersal of fragments from source populations into open sand gaps in Florida scrub, including burned areas, is likely the primary method of short-term population recovery [11]. Populations grow via vegetative propagation, or clonal growth, in which parental genotypes or thalli (genets) produce individuals (ramets) that are capable of independent growth and dispersal [22]. However, recolonization from disjunct sites is unlikely and/or infrequent, as source populations are often kilometers away. Despite the apparently strict clonal propagation in FPC, genetic variation was found to be highly structured across populations spanning their entire distribution. Based on five microsatellite loci, Yahr [21] found low genetic diversity within geographically restricted sites but unique variations in the three major ridge systems (Figure 1).

Previous translocation success of FPC [18], in addition to successful experimental translocations of the common but similar and co-occurring Jester lichen (*Cladonia leporina*) and Powder-puff lichen (*Cladonia evansii*), highlight one avenue to offset potential losses from catastrophic events and reintroduce populations. However, genetic diversity within and among FPC populations is not well known, raising questions about how these populations are related and how to protect genetic diversity and potential locally adapted genets. This lack of information limits recovery efforts for this species, e.g., can individuals from healthy, robust populations be transplanted to help restore declining/extirpated populations?

‘Big data’ is critical for accelerating lichen conservation by informing the monitoring of rare and endangered species [23]. For threatened and endangered species, reference genomes provide critical insight into genomic diversity and architecture, local adaptations and genetic rescue, inbreeding and deleterious mutations, outbreeding and introgression, phylogenomics, and the structure and function of communities [24]. Here, we provide a genome-level perspective into the mycobiont (the fungal partner) of the FPC, *Cladonia perforata*. In this study, we aimed to characterize genetic diversity at three distinct sites in southeastern Florida using whole genome sequencing. Are individual thalli in these populations genetically homogenous, or does variation occur within and among the sites? Using high-throughput sequencing, we (i) provide the first draft genomes for *C. perforata*, including both the nuclear and mitochondrial genomes, (ii) assess genetic diversity within and among populations at the level of the genome (nuclear and mitochondrial), and (iii) investigate diversity across the entire nuclear ribosomal cistron, which includes the standard DNA barcoding marker for fungi. Our study provides the first genome-scale insight into this imperiled lichen.

## 2. Materials and Methods

### 2.1. Field Sampling

As part of ongoing research into the conservation of FPC in the Atlantic Coast Ridge in southeastern Florida, U.S.A., we were interested in genetic diversity occurring in sites that might be potentially impacted by transplanting to help restore declining/extirpated populations. Collections of Florida Perforate Cladonia (FPC) were made from three distinct sites in southeastern Florida on the Atlantic Coast Ridge in January 2022: (i) Jupiter Inlet Lighthouse Outstanding Natural Area (“JILONA”), (ii) Jupiter Ridge Natural Area (“JRNA”), and (iii) The Arbors Preserve (“Arbors”) (Figure 1 and Figure 2). These sites represent only the extreme southeastern populations of FPC, and populations in the North Gulf Coast and Lake Wales Ridge were not sampled for this study.

JILONA is part of the Bureau of Land Management’s National Conservation Lands system, covering 120 acres of sand pine scrub and oak scrub habitat in Jupiter, Palm Beach County (ca. 6 mASL). Associated species include sand selaginella (*Selaginella arenicola*), scrub frostweed (*Crocanthemum nashii*), gopher apple (*Geobalanus oblongifolius*), nodding pinweed (*Lechea cernua*), partridge pea (*Chamaecrista fasciculata*), scrub palmetto (*Sabal etonia*), and rosemary (*Ceratiola ericoides*). JRNA covers 271 acres of sand pine scrub and oak scrub habitat (10–18 mASL) in Juno Beach, Palm Beach County. JRNA is part of a state-owned Conservation and Recreation Lands project managed by Palm Beach County. Associated species include rosemary (*Ceratiola ericoides*), extensive *Cladonia* beds (*Cladonia* spp.), huckleberry (*Vaccinium membranaceum*), gopher apple (*Geobalanus oblongifolius*), nodding pinweed (*Lechea cernua*), saw palmetto (*Serenoa repens*). The final site, the Arbors, covers 72 acres of sand pine scrub and oak scrub habitat (10 mASL) on private land and is located behind the Arbors Subdivision off Ralph Fair Road in Hobe Sound, Martin County. Associated species include rosemary (*Ceratiola ericoides*), saw palmetto (*Serenoa repens*), gopher apple (*Geobalanus oblongifolius*), and huckleberry (*Vaccinium membranaceum*). The two southernmost sites, JILONA and JRNA, are separated by the Loxahatchee River and 4.3 km. The Arbors is ca. 20 km north of JILONA and JRNA.

To maximize the probability of sampling distinct genets, 10 FPC thalli were collected to maximize the spatial distance (tens to hundreds of meters) separating sampled thalli within each of the three natural area preserves. Each thallus was immediately placed into a bond paper lichen packet and allowed to air dry for several days before shipping. A GPS point was taken at the approximate midpoint within the preserve’s population, and photos were taken of the general area and FPC on site (Figure 2).

Letters of permission and authorization permits for collecting FPC at these sites include: permit #2996 from the Florida Dept. of Agriculture and Consumer Services Division of Plant Industry—for Permission to Harvest Endangered or Commercially Exploited Plant(s) or Plant Part(s). Within this permit is The Arbors authorization letter to collect FPC from their private preserve; permit #2997 from the Florida Dept. of Agriculture and Consumer Services Division of Plant Industry—for Permission to Harvest Endangered or Commercially Exploited Plant(s) or Plant Part(s). Within this permit is Palm Beach County’s authorization letter to collect FPC at the Jupiter Ridge Natural Area.

### 2.2. High-Throughput Sequencing and Genome Assembly

Limited genetic sequence data are currently available for FPC: two sequences representing the standard barcoding marker for fungi, the ITS (GenBank accession numbers AY753584 and AF457903), and a single sequence representing a fragment of the nuclear beta-tubulin gene (GenBank accession No. AF458570). To generate genome-scale resources for this study and future research, a single specimen, ‘Arbors7’ (Brigham Young University Herbarium of Non-Vascular Cryptogams, Provo, UT, USA), from the Arbors, was selected for long-read high throughput shotgun sequencing using Oxford Nanopore Technologies (ONT) MinION platform. For this specimen, DNA was extracted from three different portions of the same thallus using the E.Z.N.A. Plant DNA DS Mini Kit (Omega Bio-Tek, Inc., Norcross, GA, USA) following the manufacturer’s protocol. The ONT sequencing libraries were prepared using the ligation sequencing kit SQK-LSK114 with the Native Barcoding Kit 24 V14 (SQK-NBD114.24). The three barcoded libraries were sequenced on the MinION Mk1C system with an R10.4.1 flow cell (ONT).

Raw sequence FAST5 signals were called with Guppy v6.4.6 (ONT) using the dna_r10.3_450bps_hac.cfg model. The ONT sequencing generated 12.6 Gb from 3,735,355 reads, with an average length of 2407 bp. Nanopore reads were assembled using Flye v.2.9 [25]. The Flye assembly was further polished with the filtered Illumina reads (see below) following the Trycycler user manual using Polypolish v. 0.5.0 [26] and POLCA v. 4.0.5 [27].

The final nuclear genome sequence assembly was annotated using funannotate v1.8.7 (https://github.com/nextgenusfs/funannotate, accessed on 3 April 2023). Contigs were sorted by size, and simple repeats were identified, resulting in 1,263,743 (3.76%) masked nucleotides. Ab initio gene predication was conducted with annotations from *Cladonia grayi* Cgr/DA2myc/ss v2.0, *Lobaria pulmonaria* Scotland reference genome v1.0, *Usnea florida* ATCC 18,376 v1.0, and *Xanthoria parietina* 46-1-SA22 v1.1 and the Benchmarking Universal Single-Copy Orthologs (BUSCO) with the seed species set to *Aspergillus fumigatus* as training datasets. Biosynthetic gene clusters were annotated using antiSmash v6.0 for fungi, with all options enabled for the most thorough search possible [28].

To characterize diversity in FPC at multiple genetic scales, sequence fragments from the polished assembly were investigated to identify sequences representing (i) the lichen-forming fungus (LFF) nuclear genome, including BUSCO loci, (ii) the LFF nuclear ribosomal cistron, including the internal transcribed spacer region (ITS), and (iii) the LFF mitochondrial genome. To remove sequences from the assembly that did not represent the targeted LFF, we used a Diamond v0.9.32 blastx search against the NCBI nonredundant protein database to infer the taxonomic identity of each sequence [29]. We also compared the coverage of Illumina short reads (see below) mapped to the polished sequences to identify sequences with similar coverage and corroborate the taxonomic assignments. The LFF mitochondrial genome was identified from the assembly using BLAST comparisons [30] to known *Cladonia* spp. mitochondrial genomes. The mitochondrial genome was annotated using GeSeq as implemented in the CHLOROBOX platform using currently available NCBI RefSeq Lecanoromycetes mitochondrial genomes for BLAT reference sequences (https://chlorobox.mpimp-golm.mpg.de/geseq.html, accessed on 3 April 2023; [31]). LFF sequences from the polished ONT flye assembly were assessed to identify single-copy nuclear genes using the BUSCO v5.2.2 [32] and the “ascomycota_odb10” dataset for comparison.

As a byproduct of the assembly, we also identified the photobiont’s (*Asterochloris* sp.) chloroplast genome from the complete ONT flye assembly using BLAST comparisons [30] to known Trebouxiophyceae chloroplast genomes. The *Asterochloris* sp. chloroplast genome was also annotated using GeSeq implemented in the CHLOROBOX platform using currently available Trebouxiophyceae chloroplast genomes for BLAT reference sequences (https://chlorobox.mpimp-golm.mpg.de/geseq.html, accessed on 3 April 2023; [31]).

To assess genetic variation in the mycobiont partner (*Cladonia perforata*) of the FPC lichen, seven thalli from each area—JILONA, JRNA, and Arbors—were selected for Illumina shotgun sequencing. For these specimens, total genomic DNA was extracted using the E.Z.N.A. Plant DNA DS Mini Kit (Omega Bio-Tek, Inc., Norcross, GA, USA) and following the manufacturer’s protocol. We prepared total genomic DNA following the standard Illumina whole genome sequencing (WGS) library preparation process with Adaptive Focused Acoustics for shearing (Covaris), followed by an AMPure cleanup process. The DNA was then processed with the NEBNext Ultra™ II End Repair/dA-Tailing Module end-repair, together with the NEBNext Ultra™ II Ligation Module (New England Biolabs, Ipswich, MA, USA), while using standard Illumina index primers. Libraries were pooled and sequenced using the HiSeq 2500 sequencer in high output mode, by the DNA Sequencing Center at Brigham Young University, Provo, UT, USA, with 125 cycle paired-end (PE) reads.

### 2.3. Phylogenetic Datasets and Tree Inference

Three phylogenomic data matrices were created from the nuclear genome data: (i) one based on the 41 largest sequences (each fragment > 100 Kb and spanning over 32 Mb) in the polished Flye assembly, (ii) an alignment based on coding regions from the annotated nuclear genome (spanning ca. 13 Mb), and (iii) the final based on the 1250 complete, single-copy BUSCO markers. For the nuclear data matrices, we used RealPhy v1.12 [33] to align Illumina short reads from all samples to the targets implementing the following parameters: -readLength 100 -perBaseCov 5 -gapThreshold 0.5.

For the mitochondrial phylogenomic matrix, we constructed a multiple sequence alignment (MSA) from the complete mitochondrial genome. Illumina short reads from each specimen were mapped to the mitochondrial genome assembly (specimen ‘Arbors7’) using the Geneious Prime Read Mapper, with the “medium-low sensitivity/fast” settings, iterated five times [34]. The resulting consensus sequences were aligned using the program MAFFT v7 [35,36], implementing the G-INS-i alignment algorithm, and the remaining parameters were set to default values.

We also generated alignments of sequences representing the nuclear ribosomal cistron (nrDNA), which includes the standard barcoding marker for fungi, the ITS [37], the nuclear ribosomal small and large subunits, and the intergenic spacer region [38]. We identified the nrDNA cistron from the complete ONT Flye assembly using BLAST comparisons [30]. Illumina short reads were then mapped back to the reference nrDNA sequence using the Geneious Prime Read Mapper, with the “medium-low sensitivity/fast” settings, iterated five times [34]. Considering relative coverage, we selected the longest gap-free nrDNA region shared across all samples. The resulting consensus sequences were aligned using the program MAFFT v7 [35,36], implementing the G-INS-i alignment algorithm, and the remaining parameters were set to default values. Subsequently, the ITS region was extracted from the nrDNA alignment and re-aligned along with the two available ITS sequences from *C. perforata* on GenBank (accession numbers AY753584 and AF457903) using MAFFT as described above.

From each data matrix, topologies were inferred using IQ-TREE 2 [39], with 1000 ultrafast bootstrap replicates and visualized using FigTree v1.4.4 [40].

## 3. Results

### 3.1. Genomic Data

ONT long read and Illumina short read data are available under the NCBI BioProject ID PRJNA966217.

The initial ONT Flye assembly comprised 116 fragments spanning 34.4 Mb and had a mean coverage of 211. After excluding non-target sequences, the polished *Cladonia perforata* assembly (specimen ‘Arbors7’) comprised 72 fragments. It spanned 33.61 Mb with an N50 score of 1.1 Mb (describes the length of the shortest contig in the group of the longest contigs, which together represent 50% of the assembled genome) and an L50 of 12 (the smallest number of contigs that includes 50% of the total sequence assembly length) (Table 1; Appendix A). The annotated nuclear alignment is available under the NCBI BioProject ID PRJNA982265.

BUSCO statistics suggest a relatively complete assembly, with 95.1% of the BUSCO markers recovered as a complete, single copy (1250 orthologs; Appendix A), plus an additional 3% of the total BUSCO loci when including the fragmented, single copy regions. A total of 13,980 genes were predicted for the nuclear genome of *C. perforata* (Table 1). From the polished nuclear assembly, we extracted a 9.7 Kb fragment of the nuclear ribosomal cistron (GenBank Accession Nos. OQ890691-OQ890711) bounded by difficult-to-align/map intronic regions. The complete LFF mitochondrial genome was assembled, spanning 59,014 bp (GenBank accession No. OQ934048; Appendix A). We extracted and assembled ITS reads from the *Asterochloris* photobiont from the short read data, and the ITS was highly similar (99.5%) to a recently inferred, undescribed lineage ‘OUT11’ sensu [41] (results not shown). The assembled *Asterochloris* sp. (“OTU11” sensu [41]) chloroplast genome was identified by blast comparisons and spanned 218,874 bp (GenBank accession No. OQ934049; Appendix A).

### 3.2. Phylogenomic Inferences

A summary of phylogenomic data matrices is provided in Table 2.

For the nuclear genome, coverage of Illumina short read data ranged from 4× to 15× per sample (Table 3). The phylogenomic data matrix based on the largest fragments (>100 Kb) comprised 32.1 Mb aligned nucleotide position characters (Appendix A). In the nuclear REALPHY alignment, specimens Arbors5 and JILONA7 had 53.69% and 68.86% missing data, respectively, due to low coverage. All remaining specimens had less than 20% missing data, with an average of 6.82% missing data. The resulting ML topology revealed six distinct, closely related genets, with no genets shared among the sampling sites (Figure 3a), e.g., all sampled sites harbored unique genets. Phylogenies inferred from the alignment of the annotated coding regions (12.1 Mb) and the alignment of the concatenated 1250 BUSCO markers (1.9 Mb) recovered the same six genets, although relationships differed among the topologies inferred form different nuclear data matrices (Appendix A).

Three genets were sampled in the Arbors (the smallest protected area sampled), two genets were sampled in JRNA (the largest protected area sampled), and a single genet was sampled in JILONA. Two of the six genets—one from the Arbors and the second from JRNA—were represented by a single sampled thallus.

The mitochondrial genomes of all sampled thalli were 100% identical. Coverage of the mitochondrial genome with Illumina short read data ranged from 61× to 287× per sample (Table 2). Phylogenies inferred from portions of the nuclear ribosomal cistron failed to recover the same six genets inferred from the nuclear genomic data matrices but did reveal two distinct genetic groups. The topology inferred from the ITS data revealed two well-supported clades: one comprising all specimens collected from JRNA and the Arbors, and the second contained the specimens collected from JILONA, plus the only two presently available sequences from GenBank (Figure 3b). ITS sequences generated from specimens collected at JILONA were 100% identical but differed from those on GenBank (Lake Wale Ridge, central Florida [Figure 1]) at a few nucleotide position characters. The topology inferred from a larger fragment of the nuclear ribosomal cistron also recovered all specimens collected from JRNA and the Arbors in a well-supported clade and specimens collected from JILONA in a separate, distinct clade. Average pairwise similarity across a 9.8 Kb fragment of the cistron and the ITS were 99.7% and 99.5%, respectively (Table 2).

## 4. Discussion

Thirty years after the initial designation of the Florida Perforate Cladonia lichen (FPC) as an endangered species, we provide critical genomic resources to aid in conservation efforts and understanding the biology of this imperiled lichen [23,24,42]. The draft nuclear genome generated by ONT long read data, and polished using Illumina data, spans over 33 Mb and is estimated to be over 95% complete (Table 1). The total genome size assembled here is slightly smaller than previously published *Cladonia* genomes, e.g., 33.6 Mb for *C. perforata* vs. 34.6 to 37.1 Mb in other *Cladonia* species [43,44]. While recent lichen-forming fungal genomes generated using ONT long read data have resulted in nearly complete, telomere-to-telomere chromosomal assemblies [43], our nuclear genome assembly of *C. perforata* was somewhat more fragmented (Table 1), while still within the range of high-quality draft genomes—see summary in [43]. The complete, circular mitochondrial genome for *C. perforata* was 59 Kb (Table 3), falling within the upper size range of other sequenced *Cladonia* spp. mitochondrial genomes, 45,312–60,062 bp [45]. Finally, we provide the first chloroplast genome, to our knowledge, for the photobiont genus associated with FPC, *Asterochloris* sp.

Using a genome-wide data matrix spanning over 32 Mb, we inferred the presence of multiple distinct mycobiont genets occurring at local scales in southeastern Florida (Figure 3a). Strikingly, no genets were shared among even the closest sample sites (separated by less than 5 km; Figure 1), with each sampled protected area harboring unique FPC genets (Figure 3a). The same six genets were inferred using a subset of the nuclear genome—1250 BUSCO loci, spanning 1.9 Mb. Our results are partly consistent with previous inferences from microsatellite data from *C. perforata*, revealing highly structured genetic variation among populations [21]. Yahr [21] found that *C. perforata* populations in the Atlantic Coast Ridge (area sampled for this study) were genetically more similar to each other than those occurring in the North Gulf Coast and the Lake Wales Ridge regions (Figure 1). While previous work documented low within-site variation, we found multiple distinct genets occurring in two of the three sites sampled (Table 3). This suggests that the range of genetic variation in FPC throughout its entire distribution is likely much higher than that sampled here. Highly genetically structured, isolated FPC populations would be consistent with the assumption that these are a part of unique Florida scrub communities that have likely remained intact since the formation of the dunes during the Pleistocene [1,2,3]. Interestingly, earlier works suggest that isolation by distance cannot explain the diversity within and among *C. perforata* populations across its distribution [21], and other factors potentially influencing dispersal merit additional attention. Restriction site-associated DNA sequencing data and other target enrichment approaches [46,47,48,49,50] will be essential for determining evolutionary relationships and diversity, landscape genetics, and the range of clonal ramets across the entire distribution of FPC.

Data from the nuclear ribosomal cistron (ca. 10 Kb) of the mycobiont nuclear genome, including the standard fungal DNA barcoding marker (the internal transcribed spacer region) [37], also provides evidence of genetic variation in the sampled specimens, recovering two distinct groups, rather than six inferred from nuclear genome-scale data (Figure 3). The ITS marker was insufficient to distinguish at least some genets, e.g., five putative genets from the “Arbors” and Jupiter Ridge Natural Area (JRNA) were indistinguishable using ITS data, while the single genet occurring in Jupiter Inlet Outstanding Natural Area (JILONA) was distinct from all the others based on a multiple sequence alignment of the ITS (Figure 3b). Additional genetic variation was found in the ITS region when including the two presently available ITS sequences from GenBank (Figure 3b) [21]. These results suggest that the standard fungal DNA barcode (ITS) may be a reasonable “first pass” marker for screening genetic diversity in the FPC mycobiont, given the relatively inexpensive cost compared to genome-scale sequencing. Specimen sampling for the present study was restricted to a narrow portion of the overall patchy distribution of FPC (Figure 1). We predict that with broader sampling spanning distinct populations throughout the entire distribution, additional mycobiont ITS haplotypes would be identified in FPC. This is supported by the differences observed among the ITS sequences inferred from our samples in southeastern Florida and the two sequences from GenBank from specimens collected in the Lake Wales Ridge region. However, nuclear genome-scale data is required to identify genetic variation on the scale of genets (Figure 3).

In contrast to the data from the nuclear genome, the entire mitochondrial genome was identical in all sampled specimens. In contrast to many animal lineages where mitochondrial genomes have high resolving power for population genetics and phylogeography [51], mitochondrial genomes may have more limited utility for identifying population structure in FPC [47]. Intraspecific variation in mitochondrial genomes has been observed in two widespread species in *Cladonia* [52]. Extended FPC sampling beyond the Atlantic Coast Ridge (Figure 1) will be critical to determining the full range of mitochondrial variation in this lichen.

FPC is a photobiont specialist, associating with multiple geographically structured and closely related *Asterochloris* genotypes [17]. These photobiont lineages appear to have limited geographic distributions, and restricted availability of compatible *Asterochloris* photobionts may further limit the successful establishment of FPC [17]. More recently, the geographical distribution of the different *Asterochloris* lineages associating with *Cladonia* species suggests that mycobiont identity and climate are the main predictors of patterns of *Asterochloris* genetic variation [41]. Conservation efforts to restore declining/extirpated populations should consider the availability of suitable *Asterochloris* partners.

An increasing number of published landscape genetic and genomic studies of LFF show that the genetic diversity of populations and rates of gene flow vary substantially between rare and common LFF. In the 21 individuals of FPC sampled from three sites, we found substantial genetic diversity within and between sites. Yahr [21] sampled throughout the distribution of FPC and inferred the population structure using five microsatellite markers. She found that all the populations were genetically distinct from each other, suggesting low rates of gene flow among sites. A population genomics study of the only other federally endangered lichen, *Cetradonia linearis*, similarly found evidence for strong isolation by distance and for high genetic diversity within populations [53]. These findings contrast strongly with studies of widespread LFF. *Cladonia stellaris*, for instance, was found to lack spatial genetic structure across a very large geographic area in Quebec, Canada [54], and Alors et al. [55] recovered a similar pattern in *Parmelina carporrhizans*. In a circum-Antarctic study of *Pseudocyphellaria glabra,* Widhelm et al. [46] found evidence for clades that largely correspond to continental-level regions with few instances of intercontinental gene flow, thus showing some spatial structure, though on a very large spatial scale. In *Lobaria pulmonaria*, a species with a circumboreal distribution that is rare and patchily distributed in some parts of its range, there is evidence for high levels of isolation by distance in some parts of its range [56] and evidence for gene pools with very large distributions in other parts of its range [57], which is perhaps driven by differing landscape-level processes occurring among sampled regions. The hypothesized relationship between rarity and migration rates in LFFs remains to be tested in a rigorous framework. If a similar pattern is found across many rare species, it would suggest that assisted migration may be an effective conservation action for rare LFF.

In the pyrogenic Florida scrub ecosystem, gap specialists, such as FPC, are sensitive to both alterations to fire regimes and microhabitat availability [58,59]. With increasingly fragmented habitat in the Florida scrub, the impact of disturbances on isolated FPC populations becomes increasingly problematic. In these fire-dependent communities, the dispersal of FPC from undisturbed habitats into recently burned areas is likely the primary method of short-term population recovery [11]. Post-fire and post-hurricane research and reintroduction observations suggest that FPC is not only limited by dispersal but also that populations may also be slow to increase in abundance [11].

From a conservation perspective, fine-scale genetic resolution is needed to resolve questions of dispersal and genetic diversity. Our results suggest that even at local scales, i.e., sites in close geographic proximity, such as those in the Atlantic Coast Ridge sampled here, transplants will likely introduce novel genets (Figure 3a). Whether distinct genets represent local adaptations remains unknown. Recent translocation studies resulted in significant population size increases at Jupiter Inlet Lighthouse Outstanding Natural Area [18], suggesting the likely novel genets transplanted to distinct sites were able to persist and grow. Considering dependency on fire for maintaining suitable habitat for FPC, infrequent dispersal events are likely fundamental to the persistence of FPC, although genetic variation cannot be accounted for strictly through isolation by distance [11,21].

Given the imperiled state of many FPC populations, we predict that the benefit of translocations from robust populations to restore disturbed populations outweighs the risk of introducing novel genetic variation. Pending additional genetic screening, restricting translocations to populations occurring within the same general region is likely advisable, e.g., populations within the North Gulf Coast, Lake Wales Ridge, or Atlantic Coast Ridge, respectively.

The genomic resources generated here—the nuclear and mitochondrial genomes of the mycobiont *C. perforata* and the photobiont’s chloroplast genome—provide critical resources for genome-informed conservation efforts for the FPC.

## Figures and Tables

**Figure 1 jof-09-00698-f001:**
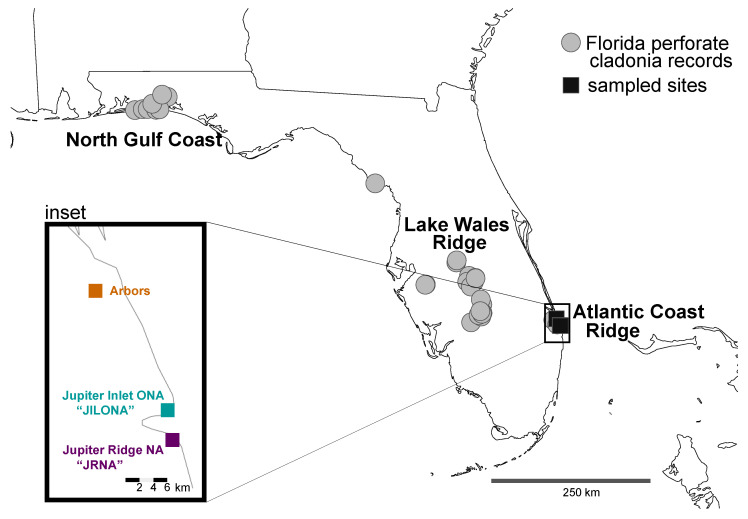
Distribution of the Florida Perforate Cladonia (FPC) in Florida, USA, based on herbarium records available on the Consortium of Lichen Herbaria (https://lichenportal.org/portal/; accessed on 24 April 2023). Grey circles indicate historic FPC records, and the black squares in southeastern Florida on the Atlantic Coast Ridge indicate the three sites sampled for this study—shown in the inset.

**Figure 2 jof-09-00698-f002:**
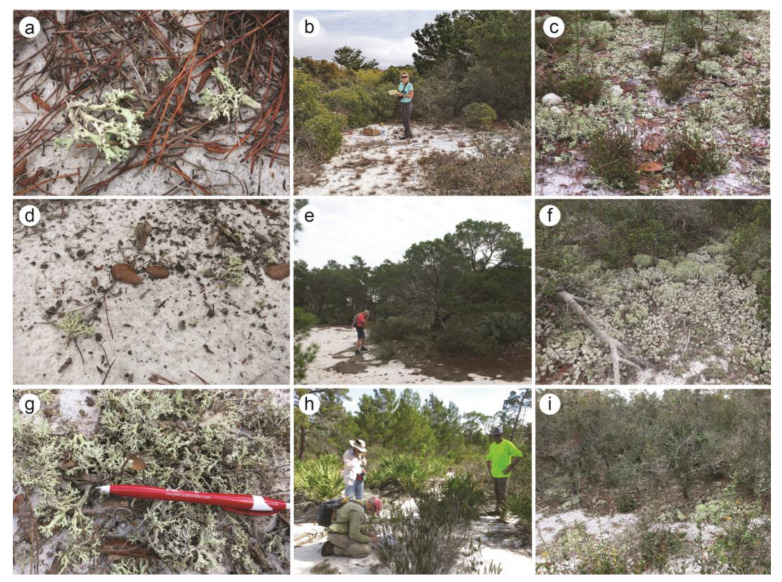
Photographs of habitat and Florida Perforate Cladonia (FPC) at the three sites sampled in southeastern Florida in 2021. (**a**–**c**): Jupiter Inlet Lighthouse Outstanding Natural Area (“JILONA”); (**d**–**f**): Jupiter Ridge Natural Area (“JRNA”); and (**g**–**i**): The Arbors Preserve (“Arbors”). Panels on the left are images of FPC habit, panels in the center depict typical habitat at the sample site, and panels on the right depict typical *Cladonia* beds. Photo credit: A. DeBolt, except panel ‘b’ from R. Rosentreter.

**Figure 3 jof-09-00698-f003:**
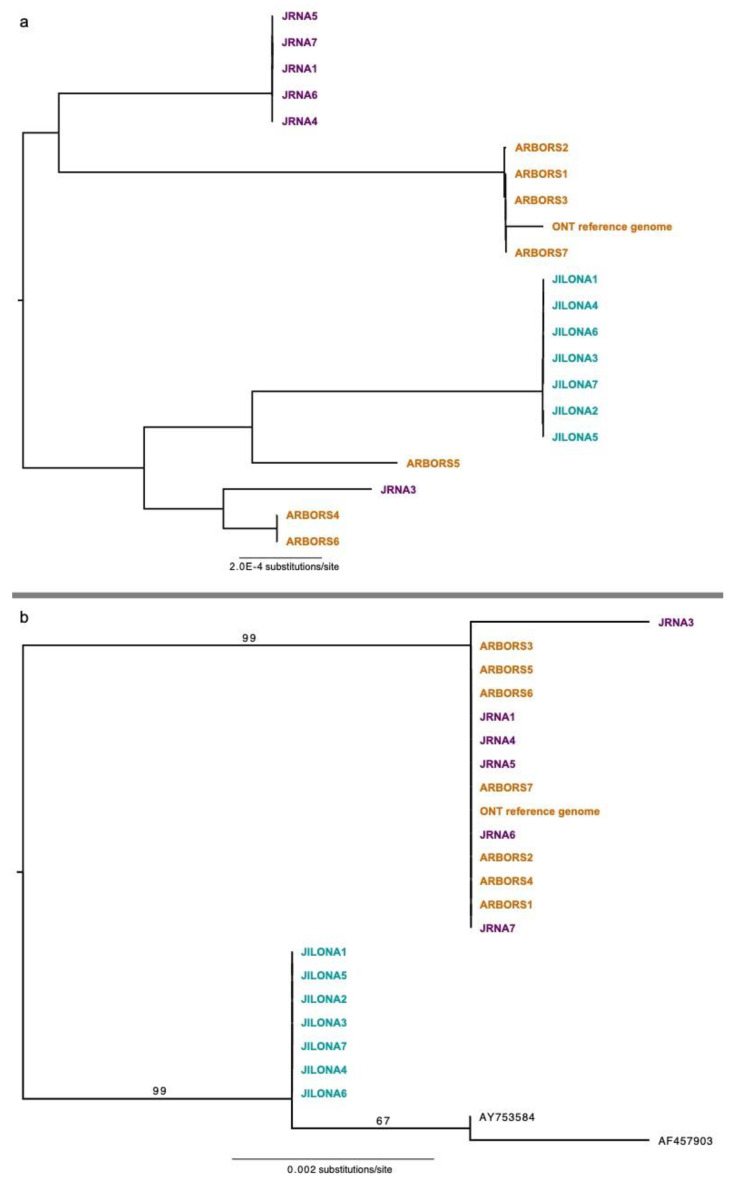
Phylogenetic inferences based on a 32.1 Mb nuclear phylogenomic matrix (panel (**a**)) and an alignment of the standard fungal DNA barcode, the ITS (panel (**b**)). Samples from the three distinct sites are color-coded: Jupiter Inlet Lighthouse Outstanding Natural Area (“JILONA”); Jupiter Ridge Natural Area (“JRNA”); and the Arbors. All relationships in panel (**a**) were recovered with 100% bootstrap support, and bootstrap values in panel (**b**) are shown above branches.

**Table 1 jof-09-00698-t001:** Summary of nuclear genome of the mycobiont of the Florida Perforate Cladonia (*Cladonia perforate*—specimen ‘Arbors7’). The size of the assembled fragments is reported, including the number of fragments > 10 kb, >100 kb, and >1 Mb. Annotations results of predicted gene categories from funannotate v. 1.8.7, including carbohydrate-active enzymes (CAZYmes), biosynthetic gene clusters (antiSMASH clusters), biosynthetic enzymes (antiSMASH biosynthetic enzymes), and secondary metabolism gene families (antiSMASH smCOGs).

# of Contigs	72
Total size	33.61 Mb
Longest/shortest contig	1.75 Mb/2 Kb
>10 kb/>100 kb/>1 Mb	53/41/14
N50/L50	1.1 Mb/12
GC content	47.05%
BUSCO complete single	1250/1315 (95.1%)
BUSCO complete duplicate	2/1315 (0.2%)
BUSCO fragmented	40/1315 (3.0%)
BUSCO missing	23/1315 (1.7%)
Total annotations	13,980
CAZYmes	215
antiSMASH clusters	48
antiSMASH biosynthetic enzymes	99
antiSMASH smCOGs	141

**Table 2 jof-09-00698-t002:** Summary of *Cladonia perforata* multiple sequence alignments. Pairwise identification was not estimated for nuclear genomic data, including coding and BUSCO loci (indicated as “NA”).

Data Matrix	MSA Size	Genetic Groups	Pairwise Identity (Identical Sites)
Nuclear Genome	32.1 Mb	6	NA
Coding loci	12.1 Mb	6	NA
BUSCO loci	1.91 Mb	6	NA
Cistron	9777 bp	2	99.7% (99.3%)
ITS	565 bp	2 (3) ^1^	99.5% (98.1%)
Mitochondrial Genome	59,014 bp	1	100% (100%)

^1^. ITS data include the two currently available *Cladonia perforata* sequences available on GenBank (accession Nos. AF457903 and AY753584); the two GenBank sequences formed a third distinct group, closely related to the sequences generated from specimens from Jupiter Inlet Outstanding Natural Area.

**Table 3 jof-09-00698-t003:** Florida Perforate Cladonia specimens sampled for Illumina high-throughput sequencing. The geographic origin for each of the three populations is coded as: “Arbors” for the Arbors Preserve, “JILONA” for Jupiter Inlet Lighthouse Outstanding Natural Area, and “JRNA” for Jupiter Ridge Natural Area. The total number of Illumina short reads per sample is reported along with the average coverage for the nuclear, mitochondrial (mt), and nuclear ribosomal cistron (nrDNA) of the mycobiont *Cladonia perforata* per sample.

Sample Code	Illumina Reads	Nuclear Coverage	mt Coverage	nrDNA Coverage
Arbors1	3,204,340	9×	169×	32×
Arbors2	5,439,708	15×	251×	50×
Arbors3	4,727,072	12×	186×	65×
Arbors4	5,475,152	14×	287×	36×
Arbors5	1,927,270	4×	61×	11×
Arbors6	3,623,778	8×	130×	35×
Arbors7 ^1^	3,646,488	10×	159×	53×
JILONA1	4,254,622	9×	187×	54×
JILONA2	4,915,262	12×	185×	64×
JILONA3	6,474,484	15×	271×	66×
JILONA4	4,708,534	12×	215×	52×
JILONA5	4,270,992	11×	175×	72×
JILONA6	4,627,850	11×	209×	36×
JILONA7	2,052,574	4×	84×	26×
JRNA1	4,086,004	9×	170×	16×
JRNA2 ^2^	4,827,906	12×	165×	94×
JRNA3	3,255,906	8×	133×	26×
JRNA4	5,457,804	13×	230×	29×
JRNA5	4,800,882	12×	225×	30×
JRNA6	3,589,860	8×	207×	18×
JRNA7	5,092,998	13×	245×	40×

^1^ Specimen ‘Arbors7’ was selected for long read de novo shotgun sequencing using the Oxford Nanopore Technologies platform. ^2^ Specimen ‘JRNA2’ represents *Cladonia leporina* and was excluded from all analyses.

## Data Availability

Oxford Nanopore Technologies long read and Illumina short read data are available under the NCBI BioProject ID PRJNA966217; the *Cladonia perforata* nuclear genome is available on GenBank under the NCBI BioProject ID PRJNA982265.; the *Cladonia perforata* mitochondrial genome is available on GenBank as accession No OQ934048; the chloroplast genome from the photobiont associated with *Cladonia perforata* specimen ‘Arbors 7’, an *Asterochloris* sp., is available on GenBank as accession No OQ934049; and all data matrices are available as supplemental files included in the published version of the manuscript.

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
