# Peer review of "Genomic Resources for the First Federally Endangered Lichen: The Florida Perforate Cladonia (Cladonia perforata)"

_jof, 2023, doi:10.3390/jof9070698_

Round 1
Reviewer 1 Report
The authors of this work are reputed and expert lichenologists and therefore the work is of high quality.
Even though the subject matter is somewhat dense, the work is easy to read and the objective and results obtained are clearly understood.
I have simply suggested some minor changes in a word version that I have transformed from the pdf.
I hope that these suggested changes will serve to increase the value of the work.
My congratulations to the authors for their paper.

Author Response
Well-written, nice paper dealing with population ecology of endemic lichen species – Cladonia perforata. Still, I would have expected more background information in Introduction. For example, that some population genetics has been done already for this species. And also, examples of other protected species for which similar studies has been done, would have been nice. Now I found all this information in Discussion, but from Discussion I would expect more synthesis of previous and your new results.
Response: Dr. Rebecca Yahr’s thesis includes a fantastic chapter on genetic diversity on clonal population structure in Cladonia perforata using microsatellite markers. While the study provides critical insight into population structure in C. perforata, it was not peer reviewed and published beyond her dissertation. While the study was done well, in my opinion, we attempted to walk the line between recognizing this critical research and avoid citing unpublished research. We’ve made a few modifications in the Introduction to try to better recognize Yahr’s important work.
New text: “Despite the apparently strict clonal propagation in FPC, genetic variation was found to be highly structure across populations spanning their entire distribution. Based on five microsatellite loci, Yahr [21] found low genetic diversity within geographically restricted sites but unique variation in each of the three major ridge systems (Fig. 1). FPC was also found to be a photobiont specialist, associating with multiple geographically structured and closely related Asterochloris genotypes [17].”
From the Introduction or Material and Methods, it remains unclear for me, why you chose only these three populations which according to the map are quite close to each other and not those that are farther. Are these distant populations still there or are they destroyed? Or is this study a part of the larger project?
Modified text: “As part of ongoing research into conservation of FPC in the Atlantic Coast Ridge in southeastern Florida, we were interested in genetic diversity occurring in sites potentially impacted by transplanting to help restore declining/extirpated populations. Collections of Florida Perforate Cladonia (FPC) were made from three distinct sites in southeastern Florida on the Atlantic Coast Ridge in January 2022: (i) Jupiter Inlet Lighthouse Out-standing Natural Area (“JILONA”), (ii) Jupiter Ridge Natural Area (“JRNA”), and (iii) The Arbors Preserve (“Arbors”) (Figs. 1 and 2). These sites represent only the extreme southeastern populations of FPC, and populations in the North Gulf Coast, and Lake Wales Ridge (Fig. 1) were not sampled for this study.”
Discussion. Considering the propagation mode, and compared with other LFFs with different propagation mode, were your results expectable? Has anyone checked photobiont? As FPC is clonal, the photobiont should be the same throughout the area. But is it true here?
New text: "FPC is a photobiont specialist, associating with multiple geographically structured and closely related Asterochloris genotypes [17]. These photobiont lineages appear to have limited geographic distributions, and restricted availability of compatible Asterochloris photobionts may further limit the successful establishment FPC [17]. More recently, geographical distribution of the different Asterochloris lineages associating with Cladonia species suggests that mycobiont identity and climate are the main predictors of patterns of Asterochloris genetic variation [41]. Conservation efforts to restore declining/extirpated populations should consider availability of suitable Asterochloris partners."
Materials and methods,
Line 65, legend of Figure 1: It will be nice if you use the same abbreviations for sites as for samples. It will be easier to bring together the results (e.g. Fig. 3) and the position of the sites.
We’ve revised the figure to include the same site abbreviations used in the subsequent figures.
Line 106: also refer Figure 1 here.
Added
Line 113: delete double dots.
Done
2.2. High ...
Corrected
- 159 – reference to Guppy is missing
Guppy is an Oxford Nanopore Technology (ONT) program, and we indicate that it is a part of ONT.
- 178 – reference to Diamond v.0.9.32 is missing
Added
- 204 - New England Biolabs – company of USA?
Added “USA”
Results.
- Table 2 (line 265). The legend is somewhat incomplete. Does NA mean not available? not applicable?
Modified caption: “Table 2. Summary of Cladonia perforata multiple sequence alignments. Pairwise identify was not estimated for nuclear genomic data, including coding and BUSCO loci (indicated as ‘NA”).”
- L. 273 - specimens A5 and J7? You use here different abbreviations than in the rest of the text.
We’ve revised the text to match the rest of the manuscript.
- Ls. 295-296. You say that “The mitochondrial genomes of all sampled thalli were 100% identical (Table 3)”. I can’t derive that information from Table 3. And I also do not catch the following information from Table 2 - “Coverage of the mitochondrial genome with Illumina short read data ranged from 61× to 287× per sample (Table 2)”
We’ve deleted the reference to Table 2 from this sentence.
Discussion
- Ls. 360-361: You say “…while the single genet occurring in Jupiter Inlet Outstanding Natural Area was distinct from all the others based on a multiple sequence alignment of the ITS (Fig. 3b).” It is somewhat difficult to follow this train of thoughts because you do not use abbreviation(s) as in Figure 3.
Revised text: “The ITS marker was insufficient to distinguish at least some genets, e.g., five putative genets from the “Arbors” and Jupiter Ridge Natural Area (JRNA) were indistinguishable using ITS data, while the single genet occurring in Jupiter Inlet Outstanding Natural Area (JILONA) was distinct from all the others based on a multiple sequence alignment of the ITS (Fig. 3b).”
- Lines: 379-381: You say that “An increasing number of published landscape genetic and genomic studies of LFF show that genetic diversity of populations and rates of gene flow vary substantially between rare and common LFF.” I guess there may be some studies correlating rarity vs. commonness pattern with reproduction mode, photobiont availability, etc. You can do some generalizations here.
Given the limited number of studies, we are hesitant to provide generalization here.
References
The style of the references should be unified according to the rules of the journal. In the current form, it is somewhat sloppy.
We have carefully revised the citation list to match the journal style.
Reviewer 2 Report
Well-written, nice paper dealing with population ecology of endemic lichen species – Cladonia perforata. Still, I would have expected more background information in Introduction. For example, that some population genetics has been done already for this species. And also, examples of other protected species for which similar studies has been done, would have been nice. Now I found all this information in Discussion, but from Discussion I would expect more synthesis of previous and your new results.
From the Introduction or Material and Methods, it remains unclear for me, why you chose only these three populations which according to the map are quite close to each other and not those that are farther. Are these distant populations still there or are they destroyed? Or is this study a part of the larger project?
Discussion. Considering the propagation mode, and compared with other LFFs with different propagation mode, were your results expectable? Has anyone checked photobiont? As FPC is clonal, the photobiont should be the same throughout the area. But is it true here?
Materials and methods,
Line 65, legend of Figure 1: It will be nice if you use the same abbreviations for sites as for samples. It will be easier to bring together the results (e.g. Fig. 3) and the position of the sites.
Line 106: also refer Figure 1 here.
Line 113: delete double dots.
2.2. High ...
l. 159 – reference to Guppy is missing
l. 178 – reference to Diamond v.0.9.32 is missinf
l. 204 - New England Biolabs – company of USA?
Results.
· Table 2 (line 265). The legend is somewhat incomplete. Does NA mean not available? not applicable?
· L. 273 - specimens A5 and J7? You use here different abbreviations than in the rest of the text.
· Ls. 295-296. You say that “The mitochondrial genomes of all sampled thalli were 100% identical (Table 3)”. I can’t derive that information from Table 3. And I also do not catch the following information from Table 2 - “Coverage of the mitochondrial genome with Illumina short read data ranged from 61× to 287× per sample (Table 2)”
Discussion
· Ls. 360-361: You say “…while the single genet occurring in Jupiter Inlet Outstanding Natural Area was distinct from all the others based on a multiple sequence alignment of the ITS (Fig. 3b).” It is somewhat difficult to follow this train of thoughts because you do not use abbreviation(s) as in Figure 3.
· Lines: 379-381: You say that “An increasing number of published landscape genetic and genomic studies of LFF show that genetic diversity of populations and rates of gene flow vary substantially between rare and common LFF.” I guess there may be some studies correlating rarity vs. commonness pattern with reproduction mode, photobiont availability, etc. You can do some generalisations here.
References
The style of the references should be unified according to the rules of the journal. In the current form, it is somewhat sloppy.
Author Response

(The authors gave the same response as above.)

Reviewer 3 Report
overall comments
The manuscript provides much needed genomic data for the endangered lichen species Cladonia perforata. The dataset will benefit future conservations effort for this particular species. The manuscript was generally well written with some areas that can use more explanations/clarifications to make the manucript more complete. Some of the comments might be the results of me not being very familiar with all the genomic pipelines here. My apology in advance if some of the comments might be trivial for genomic analyses.
specific comments
- L. 20-23: The statements about multiple distinct genets and the identical mitochondrial genome+low variability in nuclear ribosomal cistron feels a bit contradictory (not that they are). Maybe it would be clearly if we can attribute the cause of distinct genets to particular part of (nuclear) genomes?
- L. 86: "Big data" here might be a bit too broad for this particular purpose (but highly marketable, understandably). "Genomic data" might be more specific?
- L. 100: The objectives here did not mention the generation of the chloroplast genome of the associated photobiont (which was mentioend in the abstract). Was that an unintended by-product of this study?
- L. 102: While it is understanable that genomic sequencing can be costly prohibitive and thus limits the number of sample, it would be nice to have an explanation/justification as to why the authors focus on this particular area on Atlantic Coast Ridge, instead of trying to cover the other disjunct populations. i.e. why sampling from three sites in a relatively small area, instead of sampling three disjunct sites to represent the species.
- L. 170: The full name of BUSCO should first appear here.
- L. 195: missing a close parenthesis here.
- L. 260 (Table 1). Some of the summary rows are not clear to the general readers, such as ">10kb/>100kb/>1Mb", "N50/L50", "antiSMASH smCOGs" etc. Make sure that these abbrevations are clearly explained somewhere in the table.
- L. 265: Why are the pairwise identty fro nuclear genome, coding loci, BUSCO loci "NA"?
- L. 311: (Figure 3) it wasn't clear from the text or the caption what the "ONT reference genome" in these phylogenies are. Is it the long-read version of ARBORS7?
- L. 317: This might be out of scope for this particular manuscript, but it would be nice to see some discussion regarding how the genome of this species differs from the other species of Claodnia sequenced so far (in terms of gene composition etc.).
- L. 329-331: I'm not sure I see the actual report of Asterochloris genome assembly in the manuscript so far.
- L. 363-5: It is not very clear if ITS would make a good "first pass" for genetic diversity, given that it underestimates the number of genets when compared to the genomic data.
- L. 426: It would be nice to read more about how these newly generated nuclear and mitochondrial genomes of this specie can help advance the conservation effort i.e. what kind of future data and/or analyses can be produced from this draft genome that can help us with the conservation effort of the species?
- L. 467. make sure that the GenBank accession for Cladonia perforata nuclear genome is available by the time of publication.
reference
- several species names were not italicized in the references (ref. no. 8, 43, 44, 51, 54, 57)
- capitalization is not consistent throughout the reference (ref. no14, 56)
- misspelling the species names in Ref. no.20.
- Some journal names are abbreviated while the others are not. (ref. no.12, 23).
- DOIs are not consistently presented (ref. 20,24,29,44,47,51,52,55,56 as a link, while the others are as numbers).
- Repeated DOI in ref.46.
Author Response
The manuscript provides much needed genomic data for the endangered lichen species Cladonia perforata. The dataset will benefit future conservations effort for this particular species. The manuscript was generally well written with some areas that can use more explanations/clarifications to make the manucript more complete. Some of the comments might be the results of me not being very familiar with all the genomic pipelines here. My apology in advance if some of the comments might be trivial for genomic analyses.
specific comments
L. 20-23: The statements about multiple distinct genets and the identical mitochondrial genome+low variability in nuclear ribosomal cistron feels a bit contradictory (not that they are). Maybe it would be clearly if we can attribute the cause of distinct genets to particular part of (nuclear) genomes?
Modified text: "All sampled thalli shared identical mitochondrial genomes, while the nuclear ribosomal cistron showed limited variability – highlighting the genetic resolution provided by nuclear genome-scale datasets."
L. 86: "Big data" here might be a bit too broad for this particular purpose (but highly marketable, understandably). "Genomic data" might be more specific?
"Big data" is a broad umbrella term that includes genome-scale data, but also other types of information, including ecological, physiological, etc. Here, our study focuses specifically on using genome-scale data.
L. 100: The objectives here did not mention the generation of the chloroplast genome of the associated photobiont (which was mentioend in the abstract). Was that an unintended by-product of this study?
Yes, this was a byproduct of the study. While it appears that the Florida Perforate Cladonia interacts with a closely related group of Asterochloris spp., there likely is not a single Asterochloris sp. that acts as a photobiont for the Florida Perforate Cladonia.
Modified text: "As a byproduct of the assembly, we also identified the photobiont’s (Asterochloris sp.) chloroplast genome from the complete ONT flye assembly using BLAST comparisons [30] to known Trebouxiophyceae chloroplast genomes."
L. 102: While it is understanable that genomic sequencing can be costly prohibitive and thus limits the number of sample, it would be nice to have an explanation/justification as to why the authors focus on this particular area on Atlantic Coast Ridge, instead of trying to cover the other disjunct populations. i.e. why sampling from three sites in a relatively small area, instead of sampling three disjunct sites to represent the species.
Modified text: "As part of ongoing research into conservation of FPC in the Atlantic Coast Ridge in southeastern Florida, we were interested in genetic diversity occurring in sites that might be potentially impacted by transplanting to help restore declining/extirpated populations. Collections of Florida Perforate Cladonia (FPC) were made from three distinct sites in southeastern Florida on the Atlantic Coast Ridge in January 2022: (i) Jupiter Inlet Lighthouse Outstanding Natural Area (“JILONA”), (ii) Jupiter Ridge Natural Area (“JRNA”), and (iii) The Arbors Preserve (“Arbors”) (Figs. 1 and 2). These sites represent only the extreme southeastern populations of FPC, and populations in the North Gulf Coast, and Lake Wales Ridge were not sampled for this study."
L. 170: The full name of BUSCO should first appear here.
Corrected.
L. 195: missing a close parenthesis here.
Corrected.
L. 260 (Table 1). Some of the summary rows are not clear to the general readers, such as ">10kb/>100kb/>1Mb", "N50/L50", "antiSMASH smCOGs" etc. Make sure that these abbrevations are clearly explained somewhere in the table.
We've added additional descriptions to the Table caption.
L. 265: Why are the pairwise identty fro nuclear genome, coding loci, BUSCO loci "NA"?
Because of missing data in the nuclear genomic datasets, we did not calculate pairwise similarity.
L. 311: (Figure 3) it wasn't clear from the text or the caption what the "ONT reference genome" in these phylogenies are. Is it the long-read version of ARBORS7?
Current text: "In the nuclear REALPHY alignment, specimens Arbors5 and JILONA7 had 53.69% and 68.86% missing data, respectively, due to low coverage. All remaining specimens had less than 20% missing data, with an average of 6.82% missing data. The resulting ML topology revealed six distinct, closely related genets, with no genets shared among the sampling sites (Fig. 3a), e.g., all sampled sites harbored unique genets"
and
"The topology inferred from the ITS data revealed two well-supported clades: one comprising all specimens collected from JRNA and the Arbors, and the second contained the specimens collected from JILONA, plus the only two presently available sequences from GenBank (Fig. 3b)."
L. 317: This might be out of scope for this particular manuscript, but it would be nice to see some discussion regarding how the genome of this species differs from the other species of Claodnia sequenced so far (in terms of gene composition etc.).
Yes, we agree that comparing genomic architecture among different Cladonia species would be fantastic. However, here we focused on providing genomic resources for aiding in conservation of the Florida Perforate Cladonia.
L. 329-331: I'm not sure I see the actual report of Asterochloris genome assembly in the manuscript so far.
Modified text from the Results: "We extracted and assembled ITS reads from the Asterochloris photobiont from the short read data, and the ITS was highly similar (99.5%) to a recently inferred, undescribed lineage ‘OUT11’ sensu [41] (results not shown). The assembled Asterochloris sp. (“OTU11” sensu [41]) chloroplast genome was identified by blast comparisons and spanned 218,874 bp (GenBank accession No. OQ934049; supplementary files S5 & S6)."
L. 363-5: It is not very clear if ITS would make a good "first pass" for genetic diversity, given that it underestimates the number of genets when compared to the genomic data.
Modified text: "Our data suggest that the standard fungal DNA barcode (ITS) may be a reasonable “first pass” marker for screening genetic diversity in the FPC mycobiont given the relatively inexpensive cost compared to genome-scale sequencing. Specimen sampling for the present study was restricted to a narrow portion of the overall, patchy distribution of FPC (Fig. 1). We predict that with broader sampling spanning distinct populations throughout the entire distribution, additional mycobiont ITS haplotypes would be identified in FPC. This is supported by the differences observed among the ITS sequences inferred from our samples in southeastern Florida and the two sequences from GenBank (from the Lake Wales Ridge region). However, nuclear genome-scale data is required to identify genetic variation on the scale of genets (Fig. 3)."
L. 426: It would be nice to read more about how these newly generated nuclear and mitochondrial genomes of this specie can help advance the conservation effort i.e. what kind of future data and/or analyses can be produced from this draft genome that can help us with the conservation effort of the species?
We can't anticipate the full range of possibilities for utilizing these data. Here, we focus on the most straightforward ways to utilize these data - inferring population structure across the distribution of this species. We hope that future work will use these data in creative ways to help protect this lichen.
L. 467. make sure that the GenBank accession for Cladonia perforata nuclear genome is available by the time of publication.
Revised text: "The annotated nuclear alignment is available under the NCBI BioProject ID PRJNA982265."
reference
several species names were not italicized in the references (ref. no. 8, 43, 44, 51, 54, 57)
capitalization is not consistent throughout the reference (ref. no14, 56)
misspelling the species names in Ref. no.20.
We've carefully revised the citation list.
Some journal names are abbreviated while the others are not. (ref. no.12, 23).
DOIs are not consistently presented (ref. 20,24,29,44,47,51,52,55,56 as a link, while the others are as numbers).
Repeated DOI in ref.46.
We've carefully revised the citation list.